# Dwarf and Tall Elephantgrass Genotypes under Irrigation as Forage Sources for Ruminants: Herbage Accumulation and Nutritive Value

**DOI:** 10.3390/ani11082392

**Published:** 2021-08-13

**Authors:** Rayanne Thalita de Almeida Souza, Mércia Virginia Ferreira dos Santos, Márcio Vieira da Cunha, Geane Dias Gonçalves, Valdson José da Silva, Alexandre Carneiro Leão de Mello, James Pierre Muir, Robson Elpídio Pereira Ribeiro, José Carlos Batista Dubeux

**Affiliations:** 1Department of Animal Science, Federal Rural University of Pernambuco, Dom Manoel de Medeiro Street, Dois Irmãos, Recife, Pernambuco 52171-900, Brazil; mercia.vfsantos@ufrpe.br (M.V.F.d.S.); marcio.cunha@ufrpe.br (M.V.d.C.); valdson.silva@ufrpe.br (V.J.d.S.); alexandre.lmello@ufrpe.br (A.C.L.d.M.); robsonzoot@gmail.com (R.E.P.R.); 2Federal University of Agreste of Pernambuco, Bom Pastor Avenue, Boa Vista, Garanhuns, Pernambuco 55292-270, Brazil; geanedg@yahoo.com.br; 3Texas A&M AgriLife Research, 1229 Hwy 281, Stephenville, TX 76401, USA; Jim.Muir@ag.tamu.edu; 4North Florida Research and Education Center, University of Florida, 3925 Hwy 71, Marianna, FL 32446-8091, USA; dubeux@ufl.edu

**Keywords:** chemical composition, napiergrass, irrigation, *Pennisetum*, variability

## Abstract

**Simple Summary:**

Cyclical droughts negatively impact agriculture, with deficits of water availability for the maintenance of crops destined for human food and animal production. Seasonality of forage quantity and quality is a critical obstacle to support domesticated herds over the year. Elephantgrass (*Pennisetum purpureum* Schum.) is a tropical forage widely used for feeding ruminants, mainly in the form of cut-and-carry, which has the potential to increase tropical pasture productivity, due to the large amount of roughage produced per unit of area. Research evaluated the response of tall and dwarf elephantgrass genotypes under irrigation considering its potential for complementing ruminant diets. This study showed that irrigation of elephantgrass, particularly during the dry season, may improve the regularity of forage production with good nutritive value.

**Abstract:**

This two-year study evaluated the effect of *Pennisetum purpureum* genotypes under rainfed or irrigated conditions, during the dry and rainy seasons, on herbage, leaf, and stem dry matter (DM) accumulation rates, nutritive value, and carbohydrate and protein fractionation. Treatments were tall (Iri 381 and Elefante B) or dwarf (Mott and Taiwan A-146 2.37) genotypes under rainfed or irrigated conditions. Taiwan A-146 2.37 (146 kg DM ha per day) showed similar herbage accumulation rate (HAR) to tall genotypes during the rainy season (124 and 150 kg DM/ha per day, respectively). Dwarf genotypes showed differences in leaf accumulation rate (LAR) (66 and 49 kg DM/ha per day). Mott leaf had less neutral detergent fiber (NDF) (589 g/kg DM) than Taiwan A-146 2.37 (598 g/kg DM), and tall genotypes had generally greater NDF (668 g/kg DM) than the dwarf genotypes. Irrigation increased fiber deposition in the leaf. Stems of all genotypes had lower in vitro digestible dry matter (IVDDM) (378 g/kg DM) under rainfed conditions in the rainy season. Leaf from irrigated plots had 23% more carbohydrate C fraction (160 g/kg CHO) than those from rainfed plots (122 g/kg CHO). Dwarf genotypes had generally greater nutritive value than tall genotypes. These genotypes show promise under irrigation to fill forage gaps during dry periods.

## 1. Introduction

Elephantgrass [*Pennisetum purpureum* Schum. syn. *Cenchrus purpureus* (Schumach.) Morrone] is one of the most important forage grasses in the Brazilian forage-livestock production system. This forage grass is grown in tropical, subtropical, and even in semiarid regions worldwide [1], and has been successfully used as cut-and-carry forage, silage, and under grazing conditions [2] due to its potential for herbage accumulation, especially in wetter or irrigated areas [3].

The seasonality of forage production due to seasonal variation in weather conditions affects animal output from forage-dependent livestock systems. The use of irrigation may increase productivity and reduce the seasonality of forage production and, when associated with species with large potential of potential herbage accumulation such as elephant grass, can be an important alternative to mitigate forage gaps during the dry season. Under irrigated conditions, tall elephantgrass genotypes are used, but little information is available comparing the potential of irrigated dwarf and tall elephantgrass types to provide forage with good nutritive value to reduce forage deficits. Tall and dwarf elephantgrass types generally show differences in morphological characteristics, herbage accumulation, and nutritive value of forage [4].

Mott elephantgrass is a dwarf type released as a cultivar in 1989 [5] with high forage nutritive value [6] and has been considered one of the best dwarf elephantgrasses released to date. Taiwan A-146 2.37, also a dwarf type elephantgrass, is adaptable and has stable forage production under varying environmental conditions [7]. Tall genotypes are generally more productive than dwarf types but may show less stable forage production [7] and a more rapid decline in nutritive value with maturity [8].

Tall elephantgrass genotypes show greater rates of stem elongation compared to dwarf types, which may affect canopy composition and contribute to differences in forage nutritive value [9]. However, it is important to evaluate the degree of changes in nutritive value in different plant components (e.g., leaf blade, stem) of tall versus dwarf genotypes under different management practices.

Dwarf (Mott and Taiwan A-146) and tall (Elefante B and Iri 381) elephantgrasses were evaluated by [10] in a cut-and-carry system for sheep production in the humid region of Brazil and reported that the dwarf elephantgrass cultivars Mott and Taiwan A-146 2.37 showed greater nutritive value than tall Elefante B and Iri 381. Four elephantgrass genotypes (Common, Silver, Red, and Dwarf) were evaluated by [11] in the tropical humid zone of Malaysia, who observed lower forage production for dwarf (3358 kg/ha per cut), greater leaf:stem ratio (3.18), and greater nutritive value compared with the tall genotypes.

Productivity and nutritive value of elephantgrass BRS Capiaçu was evaluated by [12] under different regrowth maturity (30, 60, 90, and 120 days) during winter season in a semiarid region of Brazil. They observed that, with increase in age, herbage mass likewise increased (760; 3999; 10,798; and 22,115 kg/ha/year, respectively) while nutritive value declined, characterized by the reduction in non-protein nitrogen (fraction A) (60, 54, 56, and 35%), crude protein (15, 11, 8, and 6%) and increase of NDF concentration (55, 63, 66, and 68%). Harvest intervals of 90 days were recommended during the winter season because the elephantgrass showed the best balance among productivity, efficiency, and nutritional value. Rapid growth, however, can contribute to increasing stem accumulation and deposition of lignified tissues in tropical forages, reducing leaf: stem ratio and forage nutritive value with maturity that can be stimulated by the presence of water [13].

Our hypothesis is that dwarf elephantgrass grown under irrigation could bridge the forage gap during dry periods by maintaining greater forage nutritive value compared to tall genotypes with progressing maturity. The objective of this study was to evaluate the differences in leaf and stem accumulation, nutritive value for ruminants, and carbohydrate and protein fractionation among tall and dwarf elephantgrass genotypes grown under rainfed or irrigated conditions during the dry and rainy seasons.

## 2. Materials and Methods

### 2.1. Experimental Site, Treatments and Experimental Design

The trial was conducted in Garanhuns, in the Agreste Meridional Region, Pernambuco, Brazil, at the Experimental Farm of the Federal Rural University of Pernambuco. The experimental site is located at 8°53′ S and 36°29′ W, 896 m above sea level. The climate is classified as tropical Aw’ according to Köppen-Geiger climate classification [14].

The soil of the experimental area was an Ultisol, and the texture was sandy clay loam. Average soil chemical characteristics were: pH (H_2_O) = 5.8; P = 4 mg/dm^3^ (Mehlich I); K = 0.27 cmolc/dm^3^; Na = 0.03 cmolc/dm^3^; Al = 0 cmolc/dm^3^; Ca = 1.8 cmolc/dm^3^; Mg = 0.4 cmolc/dm^3^; H+Al (potential acidity) = 2.5 cmolc/dm^3^; SB (sum by bases) = 2.51 cmolc/dm^3^; CEC (cation exchange capacity) = 5.01 cmolc/dm^3^; V (base saturation) = 50.1%; C = 2.1%; m = 0%; OM (organic matter) = 36.4 g/dm^3^. Weather data for the experimental period are presented in Figure 1.

The experimental design was a randomized complete block with split-plot arrangement and four replications. The main plots consisted of rainfed (non-irrigated) and irrigated, and subplots included four elephantgrass genotypes, tall (Elefante B and Iri-381), and dwarf (Taiwan A-146 2.77 and Mott) types, as subplots.

### 2.2. Plot Establishment and Management

Before planting, soil sampling was carried out using an auger hole 0–10 and 10–20 cm deep, randomly collected at three points within each subplot, totaling 96 samples. A single composite and representative soil sample of the total area was formed. After preparation of the composite samples (air-dried and sieved), they were analyzed at the Soil Laboratory of the Sugarcane Experimental Station at Carpina of the Federal Rural University of Pernambuco. The pH_H2O_, P (Mehlich I), K^+^, Na^+^, Al^3+^, Ca^2+^, Mg^2+^, H+Al, SB, CEC, V, C, and OM, were determined according to [15]. The need for soil pH correction was identified, which was carried out in accordance with regional fertilizer recommendation for elephantgrass in Pernambuco State [16], applying 500 kg/ha calcitic limestone, followed by plowing and harrowing.

After 90 days of soil amendment, the soil was sampled and chemical analysis was performed again, as described above. Those samples indicated the need to add 44 kg P/ha and 64 kg K/ha using single superphosphate and potassium chloride as sources, respectively, applied to furrows during planting.

Elephantgrass genotypes were established by vegetative propagation with 1-m spacing between rows. Each plot consisted of 546 m^2^ (91 × 6 m), with spacing of 8 m between main plots. Each subplot had 24 m^2^ (4 × 6 m), and the sampling area was 15 m^2^ (3 × 5 m). Annual maintenance soil fertilization was performed only during rainy season in a single application, following the regional fertilizer recommendation for elephantgrass in Pernambuco State [16], consisting of 100 kg N/ha and 64 kg K/ha in the planting furrow. The fertilization sources consisted of ammonium sulfate and potassium chloride, respectively.

In irrigated plots, water was applied using a drip irrigation system with approximately 95% distribution uniformity. Irrigation management distributed the water needed for restoring 100% crop evapotranspiration (ETc) based on the standardized Penman-Monteith method by FAO/56 [17]. The weather data used for ETc calculations was obtained from a weather station from the experimental site. In the first year, 8 and 179 mm of water were applied in the irrigated plots in the rainy and in the dry season, respectively. In the second year, 98 and 168 mm were applied in the irrigated plots in the rainy and dry season, respectively.

In July 2016, 90 days after planting, a staging cut at approximately 5 cm from the ground level was performed, and dead plants were replaced. For the beginning of the experimental period, a second staging cut was performed in November 2016, 120 days after replanting. Thereafter, the genotypes were subjected to successive harvests every 60 days during two consecutive years (2017 to 2018).

### 2.3. Response Variables

#### 2.3.1. Herbage, Leaf, and Stem Accumulation Rates and Nutritive Value

Two representative harvests (growing cycles) were chosen to evaluate seasonal herbage, leaf, and stem accumulation. The harvest that occurred in August and December represented the rainy and dry season, respectively, over 2 years, giving a total of 4 harvests, and results were considered as the average of harvests in each season. At each harvest, the herbage mass in each subplot was quantified by harvesting the forage at approximately 5 cm from the ground level. The forage harvested from 15 m^2^ of each subplot was weighed fresh in the field, and a subsample of five randomly selected basal tillers was collected. Each subsample was separated into leaf (blade), stem (stem plus leaf sheath), and dead material, weighed, and subsequently dried in a forced-draft oven at 55 °C until constant weight and then weighed to determine dry matter (DM) concentration. The leaf and stem weight of the subsamples was used to calculate the proportion of each component in the herbage mass, which was then used to calculate the leaf:stem ratio (L:S). Herbage mass was obtained by multiplying fresh weight of forage harvest from 15 m^2^ of each subplot by the respective DM concentration. Then, the herbage mass was divided by the number of days in the growing cycle (from the harvest of the previous cycle to the harvest of the considered cycle) to obtain the herbage accumulation rate (HAR; kg/ha per day). Leaf and stem accumulation rate (LAR and SAR; kg/ha per day) were obtained by multiplying the herbage accumulation rate by the proportion of each component (leaf and stem) divided by the regrowth duration.

The dried sub-samples were ground in a Wiley mill (MO6666, John Doe Co., Dog city, CA, USA) using a 1-mm sieve. The samples were analyzed for DM at 105 °C (930.15 method), mineral matter (MM) (942.05), ether extract (EE) (920.39), and crude protein (CP) (984.13), according to [18]. Lignin, neutral detergent fiber (NDF), and acid detergent fiber (ADF) adjusted for ash and protein (NDFap) were determined as described by [19], with modifications proposed by [20].

The in vitro digestible dry matter (IVDDM) was estimated according to [21] in a DAISY II Incubator (ANKOM^®^ Technology, Macedon, NY, USA) for 48 h with ruminal fluid and buffer solution (ruminal fermentation stage). After this period, 40 mL of HCl solution (6N) and 8 g pepsin were added, and samples were further incubated for 24 h (chemical digestion stage). The F57 bags containing the digestion residue were oven dried at 105 °C until constant weight and weighed. The experimental procedures were approved by the Ethics committee of Federal Rural University of Pernambuco (License n° 002/2020). A single rumen-fistulated cow was used as a ruminal fluid donor. This animal was fed with elephantgrass silage.

#### 2.3.2. Protein Fractionation

Non-protein nitrogen (fraction A), neutral detergent insoluble nitrogen (NDIN), and acid detergent insoluble nitrogen (ADIN) were analyzed as described by [22] and calculated according to the Cornell Net Carbohydrate and Protein System (CNCPS) as described by [23]. Protein fraction B1 + B2, composed by fractions of rapidly (B1) and intermediate (B2) rates of rumen degradation, was estimated by Equation (1):B1 + B2 = 100 − (A + B3 + C)(1)

Protein fraction B3 (insoluble fraction in the rumen) was calculated by the difference between NDIN and ADIN, and protein fraction C (insoluble in the rumen and indigestible in the small intestine) was considered as ADIN.

#### 2.3.3. Carbohydrate Fractionation

The carbohydrate fractionation was determined and calculated according to the Equations (2)–(4) described by [22]. Total carbohydrates (CHO) were calculated as noted in Equation (2).
CHO = 100 − (CP + EE + MM)(2)

Carbohydrate fractions A and B1 (which show rapid ruminal fermentation) were calculated as Equation (3):NFC = 100 − (CP + (NDFap) + EE + MM)(3)

Carbohydrate fraction B2 (carbohydrates of slow ruminal fermentation developed on cell wall) was calculated by the difference between NDFap (ash- and protein-free NDF) and fraction C. Carbohydrate fraction C (indigestible fiber) was calculated as Equation (4):C = NDF * 0.01 * LIG * 2.4/CHO(4)

### 2.4. Statistical Analysis

Data were analyzed using the Mixed Procedure of SAS University Edition. Irrigation, genotypes, seasons, and their interactions were considered fixed effects. The effects of blocks, year, and their interaction were considered random. Seasons were considered repeated measures. Averages were compared using the probability of the difference (“pdiff”) adjusted by Tukey test. Treatments were considered different when *p* ≤ 0.05. The following statistical model was used:Y*ijkl* = µ + B*i* + C*j* + D*k* + E*l* + CD*jk* + CE*jl* + DE*kl* + CDE*jkl* + ε*ijkl*(5)
where Y*ijkl* = observation, µ = population mean, B*i* = block effect (*i* = 4–1), C*j* = irrigation effect (*j* = 2–1), D*k* = genotype effect (*k* = 4–1), E*l* = season effect (*l* = 2–1), CD*jk* = effect of irrigation × genotype interaction, CE*jl* = effect of irrigation × season interaction, DE*kl* = effect of genotype × season interaction, and CDE*jkl* = effect of irrigation × genotype × season interaction and ε*ijkl* = residual error.

## 3. Results

### 3.1. Herbage, Leaf, and Stem Accumulation Rates and Nutritive Value

Herbage accumulation rate was affected by irrigation (*p* = 0.001) (Figure 2) and genotype × season interactions (*p* = 0.007) (Table 1).

Irrigation increased HAR by 44% compared to rainfed conditions (120 vs. 67 kg DM/ha per day, respectively) (Figure 2). The genotypes showed generally greater HAR during rainy season, and tall genotype Elefante B showed similar HAR compared to the dwarf types. Mott and Taiwan A-146 2.37 in turn did not differ from Iri 381, varying from 122 to 150 kg DM/ha/day (Table 1). Iri 381 showed 19% greater HAR compared to Mott during rainy season.

The SAR was not affected by irrigation (*p* = 0.19), but it was affected by genotype × season (*p* = 0.001) interaction (Table 1). The SAR was greater during the rainy season (average within genotypes ~77 kg DM/ha/day) compared to the dry season (~22 kg DM/ha/day) without differences among genotypes in the dry season. During the rainy season, Mott had the lowest SAR, while Taiwan A-146 2.37 SAR did not differ from Elefante B, which in turn did not differ from Iri 381.

There was genotype × irrigation interaction (*p* = 0.048) for LAR (Table 2) and irrigation × season (*p* = 0.005) for LAR and L:S (Table 3). It was generally greater under irrigation for all genotypes (~57 kg DM/ha/day), while under rainfed condition, genotypes did not show differences. When irrigated, Mott (66 kg DM/ha per day) had 26% greater LAR than Taiwan A-146 2.37 (49 kg DM/ha per day), but both showed no difference to Elefante B (56 kg DM/ha/day) and Iri 381 (59 kg DM/ha/day) (Table 2).

In the rainy season there was no difference in the LAR between the irrigated and rainfed conditions or between both seasons when irrigation was used (Table 3). Leaf:stem ratio was affected only by the interaction of irrigation and season (*p* < 0.0001). In the dry season, L:S ratio was 34% greater under rainfed conditions compared to irrigation. During the rainy season, however, similar L:S ratios under both irrigated and rainfed conditions were observed (Table 3).

Leaf nutritive value (Table 4) was affected by genotype × irrigation × season interaction. Greater CP (*p* = 0.005) concentrations were observed for Mott under rainfed and irrigated treatments during rainy and dry seasons compared to the other genotypes, but Taiwan A-146 2.37 did not differ from Mott, Elefante B, and Iri 381. During the dry season, CP was greater in rainfed compared to irrigated conditions for all genotypes. Mott and Taiwan A-146 2.37 IVDDM (*p* = 0.0004) was, on average, 14% greater than Elefante B and Iri 381 (Table 4). Mott NDF (*p* = 0.038) and ADF (*p* = 0.016) concentrations were generally less. Greater NDF concentrations occurred when elephantgrass genotypes were irrigated during the dry season and under irrigated, and rainfed treatments during rainy season there was no significant difference (Table 4).

During the rainy season there was no difference between irrigated and rainfed treatment forage ADF. Lower concentrations of ADF were observed in Mott and Taiwan A-146 2.37, and Elefante B and Iri 381 did not show a difference in ADF.

Leaf lignin (g/kg DM) concentration was affected by the genotype × irrigation interaction (*p* = 0.038) (Table 5). Irrigation increased lignin concentration by 29% when compared to rainfed conditions for all genotypes. Taiwan A-146 2.37 did not differ from either Mott or the tall genotypes.

Stem CP (*p* < 0.0001), ADF (*p* = 0.012), and lignin (*p* = 0.001) concentrations were affected by genotype × irrigation × season interaction (Table 6). Differences between irrigated and rainfed treatments occurred only during the dry season, with greater ADF and lignin concentrations in the stems under irrigated conditions. Taiwan A-146 2.37 showed similar stem CP compared to Elefante B under irrigated and rainfed conditions during rainy season and under irrigation during the dry season, while Iri 381 showed the lowest CP. Under rainfed conditions during dry season, Taiwan A-146 2.37 showed similar CP compared to Mott, and both were greater than Elefante B and Iri 381. Mott always contained less stem ADF and lignin concentrations than other genotypes, regardless of irrigated or rainfed conditions, while Taiwan A-146 2.37 was similar to tall genotypes. The ADF and lignin concentrations were generally less for the genotypes under rainfed conditions compared to irrigated in the dry season.

Stem NDF concentration was affected by genotype (*p* < 0.0001) (Figure 3) and irrigation × season interaction (*p* < 0.0001) (Table 7). Mott showed the lowest NDF concentration (633 g/kg DM), and Taiwan A-146 2.37 showed intermediary values (693 g/kg DM). Elefante B and Iri 381, on the other hand, showed the greatest stem NDF concentration (720 and 709 g/kg DM), respectively (Figure 3).

There was no difference in stem NDF concentration between rainy and dry season when irrigation was used, while rainfed treatments had lower stem NDF concentration during the dry season when compared to the rainy season (Table 7).

Stem IVDDM was affected by irrigation × season (*p* = 0.0002) (Table 7) and genotype × season interactions (*p* < 0.0001) (Table 8). During the rainy season, IVDDM did not show a difference between irrigated and rainfed conditions, when compared to dry season (Table 7). There was a decrease in IVDDM during the rainy season compared to the dry season (Table 8). Mott and Taiwan A-146 2.37 showed greater IVDDM compared to other genotypes and did not differ from each other (Table 8).

### 3.2. Protein Fractionation

There was an irrigation × genotype × season interaction for all protein fractionation variables from leaves and stems (Table 9 and Table 10). Leaf from Mott and Taiwan A-146 2.37 had greater proportion of N fraction A (*p* = 0.002) (average of irrigated and rainfed in rainy and dry season ~153 g/kg CP) than the other genotypes. Greater concentrations of this protein fraction (A) were observed in dry season forage in rainfed conditions, for all genotypes. When comparing rainy and dry season, lower concentrations of this fraction were observed for all genotypes during rainy season under rainfed conditions (Table 9).

No difference was observed for the B1 + B2 (*p* < 0.0001) fraction between the genotypes leaf in all conditions analyzed. Less fraction B3 (*p* < 0.0001) was observed in Mott leaf compared to other genotypes. The irrigated plots accumulated greater fraction B3 in leaf during dry season. Less indigestible N (*p* = 0.0006) fraction in leaf was observed during dry season when no irrigation was applied (Table 9).

The proportion of fraction A of CP compounds in stems (*p* = 0.001) was similar during rainy and dry season for all genotypes when irrigation was used. During the dry season, fraction A in stems was lower in irrigated conditions. Fraction B1 + B2 (*p* < 0.0001) in stems was 24% greater in rainfed conditions during dry season compared, to irrigated plants. Fraction B3 (*p* < 0.0001) of Elefante B and Iri 381 were 19% greater than Mott and Taiwan A-146 2.37 in irrigated and rainfed treatments during both seasons (Table 10). Fractions B3 and C showed lower concentrations under rainfed conditions during dry season, when compared to rainy season. Mott contained less fraction C (*p* = 0.001) proportion in stems under rainfed conditions during the rainy and dry seasons. Taiwan A-146 2.37 had similar proportions to Elefante B and Iri 381 under rainfed conditions, during dry season (Table 10).

### 3.3. Carbohydrate Fractionation

There was an interaction between irrigation × genotype for all leaf carbohydrate fractionation variables (Table 11). In general, the greatest soluble carbohydrate (A + B1) (*p* = 0.001) concentrations were detected in plants under rainfed conditions for all genotypes. Less fraction B2 (*p* < 0.0001) was observed for the Mott genotype compared to other genotypes. Fraction C (*p* < 0.0001) represents the indigestible fiber fraction, which did not differ among genotypes and increased by 23% under irrigation conditions (Table 11).

Stem A + B1 was affected only by the interaction between irrigation × genotype (*p* < 0.0001) (Table 12). Mott showed greater carbohydrate concentration for fraction A + B1 in stems under irrigation and rainfed treatments, when compared with other genotypes. Carbohydrate fraction B2 in stems was affected by genotype × irrigation interaction (*p* < 0.0001) (Table 12) and season (*p* < 0.0001) (Figure 4).

When irrigation was used, B2 fractions increased by 16%. Among the genotypes, Mott and Taiwan A-146 2.37 had fewer B2 fractions (Table 12).

Smaller B2 fractions accumulated in stems during the dry season (436 g/kg CHO) when compared with the rainy season (519 g/kg CHO) (Figure 4). The indigestible carbohydrate fraction (C) in stems was affected by genotype (*p* < 0.0001) (Figure 5A) and season (*p* = 0.006) (Figure 5B).

Mott (193 g/kg CHO) and Taiwan A-146 2.37 (203 g/kg CHO) contained a lower indigestible carbohydrate fraction than Elefante B (254 g/kg CHO) and Iri 381 (242 g/kg CHO) (Figure 5A). During the rainy season (248 g/kg CHO), there was an increase of 26% in the indigestible carbohydrate fraction compared to the dry season (183 g/kg CHO) (Figure 5B).

## 4. Discussion

### 4.1. Herbage, Leaf, and Stem Accumulation Rates and Nutritive Value

Irrigation contributed to soil-water deficit reduction during the dry season and possible water stress during rainy season due to rainfall distribution, increasing herbage accumulation rate of elephantgrass. However, when the availability of light and temperature to plant growth is reduced, the impact of irrigation can be limited [24]. This shows the influence of ambient climate on the nutritive value and morphological aspects of forage [25]. Our study suggests that when soil water availability is a limiting factor, irrigation has the potential to decrease seasonality of forage accumulation, reducing forage gap. However, during the rainy season, irrigation showed little effect on variables. This probably occurred due to favorable rainfall distribution during the rainy season, which reduced water stress.

Thus, using forage grasses with greater potential for forage accumulation, such as elephantgrass, can help to increase the efficiency of forage-based livestock systems. Additionally, during the rainy season the differences in HAR and LAR among dwarf and tall genotypes declined, indicating the potential for dwarf types with greater forage nutritive value to fill forage gaps, in line with the results found by [10] using the same genotypes in this study to feed sheep. They concluded that dwarf genotypes are recommended for cut-and-carry system, due to greater nutritive value.

Water contributes to cell elongation and plant growth [26]. Of the processes related to plant growth, cell formation (mainly cell expansion and differentiation) is the most sensitive to cell turgor because cells become turgid when water enters and cell size increases [27]. Genetic differences among genotypes contribute to differences in leaf and stem accumulation rate and affect forage nutritive value. In our study, water availability was the main factor in declining L:S ratio, due to advancing phenological phases and consequent plant maturity, that happened much faster under irrigation [28].

Similar reduction in nutritive value to our study, featured by CP and IVDDM in the dry season for the tall elephantgrass cv. ‘Roxo’ under irrigation, was reported by [3]. Reduction in CP and increase in NDF concentrations of tropical grasses in the rainy season under irrigation conditions were also reported by [29]. According to the authors, irrigation contributed to changes in grass growth, increasing SAR and reducing leaf:stem ratio. Irrigation contributed to a greater cell wall deposition, since turgor pressure increases the mechanical rigidity of cells and tissues of leaf and stems, increasing lignin concentration, which has a negative impact on IVDDM.

Water stress affects leaf and stem elongation due to increased stomatal closure and decrease in carbon dioxide and photosynthetic rate, contributing to disturbances in amino acid and carbohydrate metabolism. Thus, carbohydrates and protein metabolites such as proline and glycine accumulate in leaves and stems, in line with our finding of increased CP and soluble carbohydrates with rainfed conditions, favoring the growth and development of the ruminal microbiota [28]. Additionally, during rainy season or irrigation conditions, the high activity of meristematic regions drain most of the available assimilates, increasing stem tracheid maturation even in thin cell walls. This also contributes to secondary thickening and lignification [26], thereby influencing NDF, ADF, and lignin accumulation in elephantgrass genotypes.

According to [30], Taiwan A-146 2.37 has a spiky and erect arrangement of leaves, which probably requires more fiber to maintain, even for stems that show greater elongation capacity than Mott [9], likewise explained by the higher support requirement mentioned above. According to [31], Mott and Taiwan A-146 2.37 dwarf elephantgrasses exhibit a greater distance between the epidermis and vascular bundles, possessing a larger parenchymatic tissue area. This tissue is generally easily digested by ruminal microorganisms and probably contributed to the similar IVDDM for Mott and Taiwan A-146 2.37, despite Taiwan A-146 2.37 presenting the greater fiber concentration.

### 4.2. Protein and Carbohydrate Fractionation

Variations in CHO and CP contents were reported by [32]. The greater proportion of fraction A contributes to increased animal performance, because it is a source of readily available N important to carbohydrate fermentation and incorporation into carbon skeletons by microorganisms [33]. According to [34], grass development occurs at the expense of non-fiber carbohydrates, reducing potentially digestible nutrients and affecting forage nutritive value; it was observed in our experiment the greater that grass development under irrigated conditions was associated with increases in plant cell wall, which was probably a result of increase the thickness as the cell content (potentially digestible nutrients) decreases with the stimulation of plant growth [23,35,36], thereby reducing animal performance [33].

The carbohydrate A + B1 fraction is particularly important because it has rapid rates of rumen fermentation contributing to a greater IVDDM, because it provides the primary energy source for rumen microorganism multiplication [37].

Carbohydrate fraction C is represented by lignin, an indigestible plant fiber fraction [38]. Stems generally contain a greater proportion of vascular bundles with different proportions of lignin-rich sclerenchyma rings [39]. Resistant tissues present in the stems support plant structures (e.g., leaves and inflorescences), especially under greater water supply. Greater content of fraction C, however, contributes to reducing potentially degradable fiber fraction, positively affecting animal intake by the rumen filling effect [40]. In our study, presence of water contributed to faster plant growth and consequently increased C.

However, more research should be carried on indigestible fractions of carbohydrates, considering the advances in research and methodologies regarding the measurement of indigestible NDF. Lignin type and linkages of phenolic acids may explain a reasonable portion of the variation in indigestible NDF, but they may differ among varieties and stages of maturity, indicating that use of the same factors to explain digestibility in all forages across all agronomic conditions will likely lead to erroneous characterizations. This indicates that the laboratory detergent system and the chemical composition of the plant may not correctly describe the rate and extent of digestibility at all conditions since the linkages among fiber fractions explain only a portion of the digestion behavior [41].

Fiber accounts for most of the total carbohydrates contained in forage grasses [40], and elephantgrass is considered one of the best ruminant feed sources in warm climates [42]. Carbohydrate fractionation of various tropical grasses was evaluated by [34]. They reported greater forage accumulation rate during the rainy season, with a more rapid increase of stem fraction in the total forage mass, contributing to increases of NDF, ADF, carbohydrate fraction C, and lignin. An increase in fraction C is usually followed by the reduction in A + B1 fraction, contributing to reduced available energy for rumen microorganisms responsible for fiber carbohydrates fermentation, lowering ruminant performance.

## 5. Conclusions

Irrigation increases HAR, LAR, and SAR of elephantgrass genotypes, with the potential to minimize seasonal deficits in forage production. Taiwan A-146 2.37, one of the dwarf cultivars in our study, had similar leaf:stem ratios as tall genotypes, while dwarf type Mott, with a greater leaf:stem ratio, had the greatest forage nutritive value under irrigated and rainfed conditions. Most CHO and CP soluble fractions accumulated in leaves. Elefante B and Iri 381 had a greater proportion of C fraction, which contributed to a reduction in forage nutritive value.

Under irrigation, the advantage of the generally greater forage nutritive value of Mott dwarf elephantgrass was not limited, while Taiwan A-146 2.37 showed an intermediary nutritive value and did not differ from Mott and tall genotypes. The optimization of elephantgrass forage production potential through irrigation promoted the greatest growth rates, notably in tall genotypes. Adjustments in harvest frequency by physiological type must therefore be made in order to obtain a forage with consistent nutritive value for ruminants.

## Figures and Tables

**Figure 1 animals-11-02392-f001:**
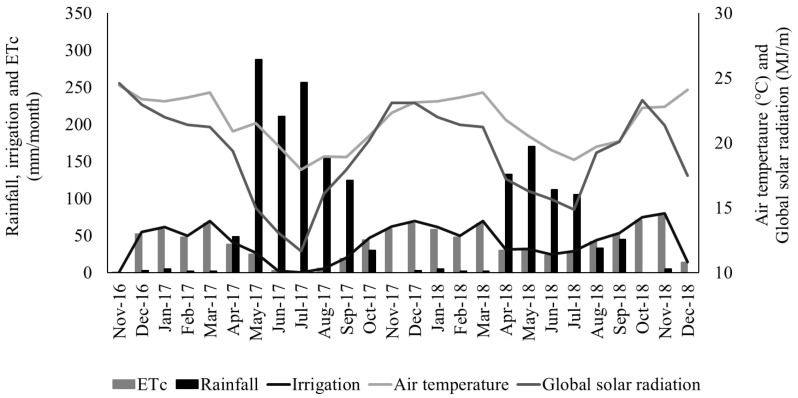
Rainfall, crop evapotranspiration (ETc), daily irrigation, air temperature, and global solar radiation recorded during the experimental period at Garanhuns, Pernambuco, Brazil.

**Figure 2 animals-11-02392-f002:**
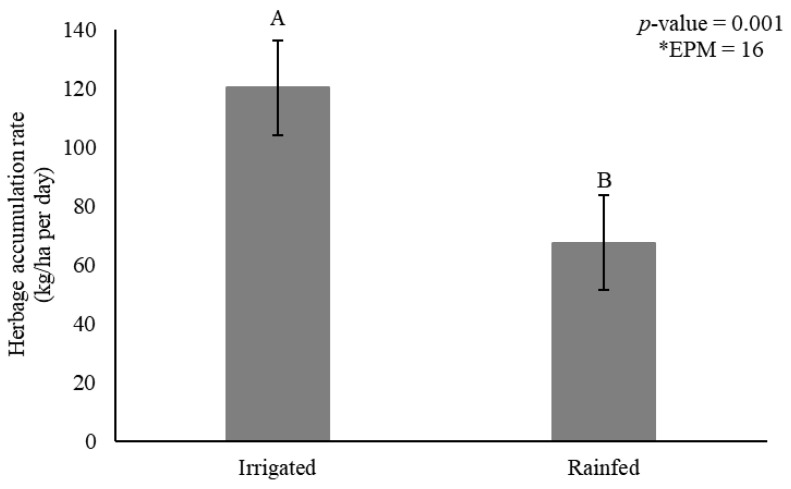
Herbage accumulation rate (HAR) (kg dry matter/ha per day) as affected by irrigation for elephantgrass genotypes at Garanhuns, Pernambuco, Brazil. *EPM = Standard error of the means.

**Figure 3 animals-11-02392-f003:**
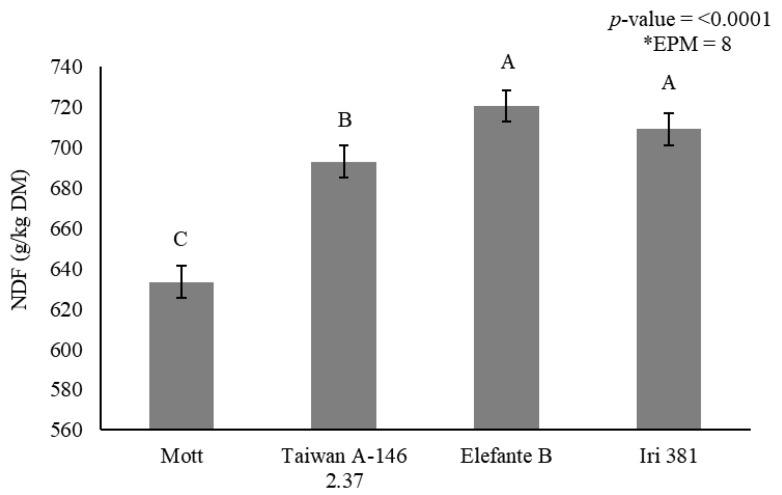
Stem neutral detergent fiber (NDF) (g/kg dry matter) as affected by genotype for elephantgrass at Garanhuns, Pernambuco, Brazil. *EPM = Standard error of the means.

**Figure 4 animals-11-02392-f004:**
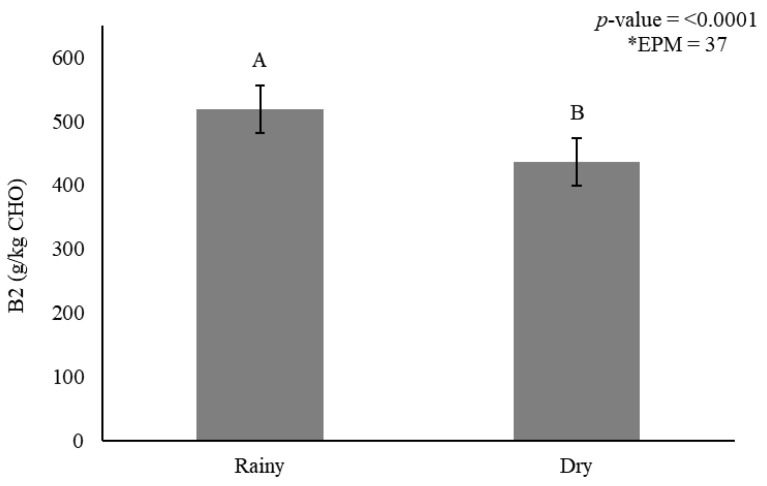
Stem carbohydrate fraction B2 (g/kg carbohydrate) as affected by season for elephantgrass genotypes at Garanhuns, Pernambuco, Brazil. *EPM = Standard error of the means.

**Figure 5 animals-11-02392-f005:**
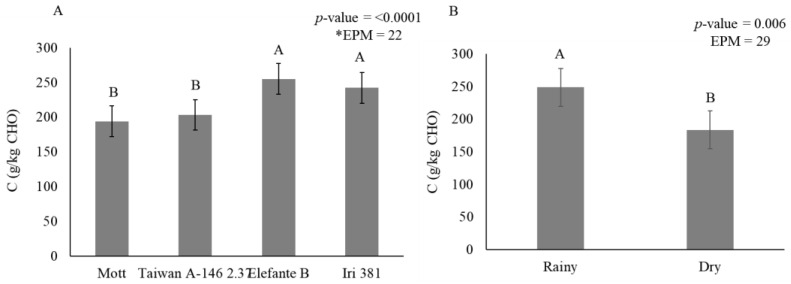
Stem carbohydrate fraction C (g/kg carbohydrate) as affected by genotype (**A**) and season (**B**) for elephantgrass genotypes at Garanhuns, Pernambuco, Brazil. *EPM = Standard error of the means.

**Table 1 animals-11-02392-t001:** Herbage accumulation rate (HAR) and stem accumulation rate (SAR; kg dry matter/ha per day), as affected by the interaction between genotype and season for elephantgrass genotypes at Garanhuns, Pernambuco, Brazil.

Genotype	Rainy	Dry	SEM ^2^	*p*-Value
HAR				
Mott	122 Ac ^1^	55 Bab	23	0.007
Taiwan A-146 2.37	146 Aab	39 Bb		
Elefante B	124 Acb	49 Bab		
Iri 381	150 Aa	63 Ba		
SAR				
Mott	57 Ac	19 Ba	8	0.001
Taiwan A-146 2.37	71 Ab	17 Ba		
Elefante B	89 Aab	22 Ba		
Iri 381	90 Aa	30 Ba		

^1^ Uppercase letters in the row compare seasons, and lowercase within a column compare genotypes, by Tukey’s test (*p* ≤ 0.05). ^2^ SEM = Standard error of the means.

**Table 2 animals-11-02392-t002:** Leaf accumulation rate (LAR; kg dry matter/ha per day) as affected by the interaction between genotype and irrigation for elephantgrass genotypes at Garanhuns, Pernambuco, Brazil.

Genotype	Irrigated	Rainfed	SEM ^2^	*p*-Value
Mott	66 Aa ^1^	36 Ba	9	0.048
Taiwan A-146 2.37	49 Ab	30 Ba		
Elefante B	56 Aab	25 Ba		
Iri 381	59 Aab	34 Ba		

^1^ Uppercase letters in the row compare irrigations, and lowercase within a column compare genotypes, by Tukey’s test (*p* ≤ 0.05). ^2^ SEM = Standard error of the means.

**Table 3 animals-11-02392-t003:** Leaf accumulation rate (LAR; kg dry matter/ha per day) and leaf:stem (L:S) ratio, as affected by the interaction between irrigation and season for elephantgrass genotypes at Garanhuns, Pernambuco, Brazil.

Season	Irrigated	Rainfed	SEM ^2^	*p*-Value
LAR				
Rainy	69 Aa ^1^	59 Aa	12	0.005
Dry	55 Aa	14 Bb		
L:S				
Rainy	0.84 Aa	0.77 Ab	0.12	<0.0001
Dry	1.32 Ba	2.01 Aa		

^1^ Uppercase letters in the row compare irrigations, and lowercase within a column compare seasons, by Tukey’s test (*p* ≤ 0.05). ^2^ SEM = Standard error of the means.

**Table 4 animals-11-02392-t004:** Leaf crude protein (CP), neutral detergent fiber (NDF), acid detergent fiber (ADF), and in vitro dry matter digestibility (IVDDM) (g/kg dry matter) as affected by the interaction between irrigation, genotype, and season for elephantgrass genotypes at Garanhuns, Pernambuco, Brazil.

Genotype	Rainy		Dry		SEM ^2^	*p*-Value
	Irrigated	Rainfed	Irrigated	Rainfed		
CP						
Mott	96 Aaα ^1^	88 Aaβ	89 Baα	140 Aaα	3	0.005
Taiwan A-146 2.37	86 Aabα	82 Aabβ	83 Babα	109 Abα		
Elefante B	73 Abα	76 Abβ	78 Bbα	107 Abα		
Iri 381	73 Abα	75 Abβ	79 Bbα	96 Abα		
NDF						
Mott	609 Abα	600 Abα	610 Abα	537 Bbβ	8	0.0038
Taiwan A-146 2.37	683 Aaα	688 Aaα	687 Aaα	626 Baβ		
Elefante B	684 Aaα	697 Aaα	686 Aaα	612 Baβ		
Iri 381	697 Aaα	670 Aaα	690 Aaα	611 Baβ		
ADF						
Mott	297 Abα	296 Abα	310 Abα	280 Bbβ	4	0.016
Taiwan A-146 2.37	320 Aaα	338 Aaα	338 Aaα	336 Aaα		
Elefante B	331 Aaα	348 Aaα	350 Aaα	332 Aaα		
Iri 381	345 Aaα	357 Aaα	324 Aabα	324 Aaβ		
IVDDM						
Mott	658 Aaα	697 Aaβ	686 Baα	726 Aaα	21	0.0004
Taiwan A-146 2.37	595 Aaα	573 Aaβ	593 Baα	695 Aaα		
Elefante B	542 Abα	548 Abβ	537 Bbα	610 Abα		
Iri 381	563 Abα	522 Abβ	555 Bbα	619 Abα		

^1^ Uppercase letters in the row compare irrigations within a season, lowercase within a column compare genotypes, and Greek letters in rows compare seasons within the irrigation treatments, by Tukey’s test (*p* ≤ 0.05). ^2^ SEM = Standard error of the means.

**Table 5 animals-11-02392-t005:** Leaf lignin (g/kg dry matter) as affected by the interaction between irrigation and genotype for elephantgrass genotypes at Garanhuns, Pernambuco, Brazil.

Genotype	Irrigated	Rainfed	SEM ^2^	*p*-Value
Mott	71 Ab ^1^	53 Bb	3	0.038
Taiwan A-146 2.37	99 Aab	64 Bab		
Elefante B	112 Aa	77 Ba		
Iri 381	116 Aa	88 Ba		

^1^ Uppercase letters in the row compare irrigations, and lowercase within a column compare genotypes, by Tukey’s test (*p* ≤ 0.05). ^2^ SEM = Standard error of the means.

**Table 6 animals-11-02392-t006:** Stem crude protein (CP), acid detergent fiber (ADF), and lignin concentration (g/kg dry matter), as affected by the interaction between irrigation, genotype, and season for elephantgrass genotypes at Garanhuns, Pernambuco, Brazil.

Genotype	Rainy		Dry		SEM ^2^	*p*-Value
	Irrigated	Rainfed	Irrigated	Rainfed		
CP						
Mott	53 Aaα ^1^	49 Aaβ	55 Baα	69 Aaα	4	<0.0001
Taiwan A-146 2.37	40 Abα	40 Abβ	47 Bbα	67 Aaα		
Elefante B	43 Abα	41 Abβ	42 Bbα	52 Abα		
Iri 381	37 Acα	32 Acβ	36 Bcα	52 Abα		
ADF						
Mott	367 Abα	360 Abα	377 Abα	308 Bbβ	10	0.012
Taiwan A-146 2.37	433 Aaα	423 Aaα	427 Aaα	362 Baβ		
Elefante B	415 Aaα	401 Aaα	416 Aaα	353 Baβ		
Iri 381	448 Aaα	434 Aaα	443 Aaα	376 Baβ		
Lignin						
Mott	112 Abα	120 Abα	119 Abα	99 Bbβ	9	0.001
Taiwan A-146 2.37	147 Aaα	146 Aaα	148 Aaα	125 Baβ		
Elefante B	145 Aaα	147 Aaα	132 Aaα	122 Baβ		
Iri 381	144 Aaα	141 Aaα	143 Aaα	129 Baβ		

^1^ Uppercase letters in the row compare irrigations within a season, lowercase within a column compare genotypes, and Greek letters in rows compare seasons within the irrigation treatments, by Tukey’s test (*p* ≤ 0.05). ^2^ SEM = Standard error of the means.

**Table 7 animals-11-02392-t007:** Stem neutral detergent fiber (NDF) and in vitro digestible dry matter (IVDDM) (g/kg dry matter), as affected by the interaction between irrigation and season for elephantgrass genotypes at Garanhuns, Pernambuco, Brazil.

Season	Irrigated	Rainfed	SEM ^2^	*p*-Value
NDF				
Rainy	723 Aa ^1^	709 Aa	10	<0.0001
Dry	706 Aa	649 Bb		
IVDDM				
Rainy	365 Aa	378 Ab	8	0.0002
Dry	364 Ba	411 Aa		

^1^ Uppercase letters in the row compare irrigations, and lowercase within a column compare seasons, by Tukey’s test (*p* ≤ 0.05). ^2^ SEM = Standard error of the means.

**Table 8 animals-11-02392-t008:** Stem in vitro dry matter digestibility (g/kg dry matter) as affected by the interaction between genotype and season for elephantgrass genotypes at Garanhuns, Pernambuco, Brazil.

Genotype	Rainy	Dry	SEM ^2^	*p*-Value
Mott	367 Ba ^1^	416 Aa	11	0.0002
Taiwan A-146 2.37	379 Ba	415 Aa		
Elefante B	317 Bb	366 Ab		
Iri 381	326 Bb	378 Ab		

^1^ Uppercase letters in the row compare seasons, and lowercase within a column compare genotype, by Tukey’s test (*p* ≤ 0.05). ^2^ SEM = Standard error of the means.

**Table 9 animals-11-02392-t009:** Leaf protein fractionation (g/kg crude protein) as affected by the interaction between irrigation, genotype, and season for elephantgrass at Garanhuns, Pernambuco, Brazil.

Genotype	Rainy		Dry		SEM ^2^	*p*-Value
	Irrigated	Rainfed	Irrigated	Rainfed		
A						
Mott	160 Aaα ^1^	170 Aaβ	160 Baα	200 Aaα	7	0.002
Taiwan A-146 2.37	130 Aabα	120 Abβ	120 Babα	170 Aabα		
Elefante B	100 Abα	90 Acβ	90 Bbα	120 Abα		
Iri 381	100 Abα	70 Acβ	70 Bbα	140 Abα		
B1 + B2						
Mott	570 Aaα	560 Aaβ	590 Aaα	600 Aaα	12	<0.0001
Taiwan A-146 2.37	570 Aaα	570 Aaα	580 Aaα	550 Aaα		
Elefante B	540 Aaα	540 Aaβ	560 Baα	590 Aaα		
Iri 381	530 Aaα	560 Aaα	570 Aaα	560 Aaα		
B3						
Mott	210 Abα	220 Abα	200 Abα	160 Bbβ	7	<0.0001
Taiwan A-146 2.37	240 Aabα	250 Aabα	240 Aaα	240 Aaα		
Elefante B	280 Aaα	290 Aaα	260 Aaα	230 Baβ		
Iri 381	280 Aaα	280 Aaα	270 Aaα	240 Baβ		
C						
Mott	60 Abα	50 Abα	50 Abα	40 Bbβ	2	0.0006
Taiwan A-146 2.37	60 Abα	60 Abα	60 Abα	40 Bbβ		
Elefante B	80 Aaα	80 Aaα	90 Aaα	60 Baβ		
Iri 381	90 Aaα	90 Aaα	90 Aaα	60 Baβ		

^1^ Uppercase letters in the row compare irrigations within a season, lowercase within a column compare genotypes, and Greek letters in rows compare seasons within the irrigation treatments by Tukey’s test (*p* ≤ 0.05). ^2^ SEM = Standard error of the means.

**Table 10 animals-11-02392-t010:** Stem protein fractionations (g/kg crude protein) as affected by the interaction between irrigation, genotype, and season for elephantgrass genotypes at Garanhuns, Pernambuco, Brazil.

Genotype	Rainy		Dry		SEM ^2^	*p*-Value
	Irrigated	Rainfed	Irrigated	Rainfed		
A						
Mott	110 Aaα ^1^	120 Aaβ	110 Baα	150 Aaα	8	0.001
Taiwan A-146 2.37	120 Aaα	110 Aaβ	110 Baα	160 Aaα		
Elefante B	80 Aaα	80 Aaβ	90 Baα	130 Aaα		
Iri 381	90 Aaα	100 Aaβ	90 Baα	140 Aaα		
B1 + B2						
Mott	360 Aaα	360 Aaβ	360 Baα	440 Aaα	7	<0.0001
Taiwan A-146 2.37	300 Aaα	290 Aaβ	350 Baα	410 Aaα		
Elefante B	230 Abα	230 Abβ	220 Bbα	360 Abα		
Iri 381	260 Abα	240 Abβ	250 Bbα	340 Abα		
B3						
Mott	410 Abα	430 Abα	420 Abα	330 Bbβ	6	<0.0001
Taiwan A-146 2.37	450 Abα	450 Abα	400 Abα	340 Bbα		
Elefante B	550 Aaα	540 Aaα	560 Aaα	390 Baβ		
Iri 381	500 Aaα	520 Aaα	510 Aaα	420 Baβ		
C						
Mott	120 Aaα	90 Abα	110 Aaα	80 Bbβ	2	0.001
Taiwan A-146 2.37	130 Aaα	150 Aaα	140 Aaα	90 Babβ		
Elefante B	140 Aaα	150 Aaα	130 Aaα	120 Baβ		
Iri 381	150 Aaα	140 Aaα	150 Aaα	100 Baβ		

^1^ Uppercase letters in the row compare irrigations within a season, lowercase within a column compare genotypes, and Greek letters in rows compare seasons within the irrigation treatments, by Tukey’s test (*p* ≤ 0.05). ^2^ SEM = Standard error of the means.

**Table 11 animals-11-02392-t011:** Leaf carbohydrate fractionation (g/kg carbohydrate) as affected by the interaction between genotype and irrigation for elephantgrass genotypes at Garanhuns, Pernambuco, Brazil.

Genotype	Irrigated	Rainfed	SEM ^2^	*p*-Value
A + B1				
Mott	390 Ba ^1^	460 Aa	6	0.001
Taiwan A-146 2.37	330 Bb	420 Ab		
Elefante B	335 Bb	400 Ab		
Iri 381	360 Bb	410 Ab		
B2				
Mott	460 Ab	430 Bb	7	<0.0001
Taiwan A-146 2.37	500 Aa	450 Bab		
Elefante B	505 Aa	480 Ba		
Iri 381	480 Aa	460 Ba		
C				
Mott	150 Aa	110 Ba	1	<0.0001
Taiwan A-146 2.37	170 Aa	130 Ba		
Elefante B	160 Aa	120 Ba		
Iri 381	160 Aa	130 Ba		

^1^ Uppercase letters in row compare irrigations, and lowercase within a column compare genotypes, by Tukey’s test (*p* ≤ 0.05). ^2^ SEM = Standard error of the means.

**Table 12 animals-11-02392-t012:** Stem carbohydrate fractions A + B1 and B2 (g/kg carbohydrate) as affected by the interaction between genotype and irrigation for elephantgrass genotypes at Garanhuns, Pernambuco, Brazil.

Genotype	Irrigated	Rainfed	SEM ^2^	*p*-Value
A + B1				
Mott	262 Ba ^1^	353 Aa	5	<0.0001
Taiwan A-146 2.37	217 Bb	270 Ab		
Elefante B	218 Bb	253 Ab		
Iri 381	218 Bb	253 Ab		
B2				
Mott	505 Ab	415 Bb	13	<0.0001
Taiwan A-146 2.37	491 Ab	409 Bb		
Elefante B	579 Aa	483 Ba		
Iri 381	564 Aa	491 Ba		

^1^ Uppercase letters in row compare irrigations, and lowercase within a column compare genotypes, by Tukey’s test (*p* ≤ 0.05). ^2^ SEM = Standard error of the means.

## Data Availability

The data presented in this study are available on request from the corresponding author.

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
