# Peer review of "Dwarf and Tall Elephantgrass Genotypes under Irrigation as Forage Sources for Ruminants: Herbage Accumulation and Nutritive Value"

_animals, 2021, doi:10.3390/ani11082392_

Round 1
Reviewer 1 Report
The introduction does not summarize the progress of domestic and international research well, and the logic is slightly confusing, so we suggest the authors to reorganize the language of this part. The experimental design has some details that need to be given in detail. Please check the correctness of the data in the results and add the missing data. The discussion repeats the results extensively, and the reasons for this result should be analyzed in depth.

Author Response
Please see the attachment the responses to the reviewer’s comments.

Reviewer 2 Report
Animals-1229387 - Dwarf and tall elephantgrass genotypes under irrigation as forage source for ruminants: herbage accumulation and nutritive value
General comments
This is an interesting paper with valuable information regarding the effect of irrigation, season and genotype of elephant grass on production and nutritive value. However, I have some issues with how the paper is written and how the data is presented, particularly in the results section. The results section and tables must be improved to make it easier for the reader to read the paper.
Specific comments
L21: The use of the word “properties” is slightly unclear to me. I presume you mean farms? Could you change properties to make this clearer?
L23-24: “This study showed that the irrigation of elephant grass, in particular during the dry season, is able…”
L26: Abstract in general; I am not sure about the exact rules for this Journal but in the abstract, you may not need to define certain variables e.g. HAR and LAR as they are only used once in the abstract. Please check this.
L34: Replace “entries” with “genotypes”
L35-36: “Under rainfed conditions in the rainy season, stems of all genotypes had lower in vitro digestible dry matter (378 g/kg).
L38: Remove “ for ruminants” from this sentence.
L58: Replace “great” with “high”
L59: “Taiwan A-146 2.37, also a dwarf type elephant grass, has been…”
L64: “elongation of tall genotypes…”
L98-99. This sentence is a bit confusing to me, particularly the phrase “and to increase base saturation to 60%”. Can you please rephrase the sentence and clarify what you mean by increasing the base saturation to 60%?
L103: Remove Pennisetum purpureum from this sentence as you have given the Latin name for elephant grass already in L52.
L124-125: Just to be clear, you have two harvests (representing a harvest in the dry and rainy season) over 2 years, giving a total of 4 harvests that were analysed for this study, is that correct? Maybe you should state that in a short sentence and state that the results are the average of the both harvests in each season?
L125-133: It is not clear what the unit of herbage mass is. This should be stated and the full methodology for determining and calculating herbage mass should be given. Further clarification on how HAR, LAR and SAR were calculated is also required. I presume that you measured herbage mass at the specific harvests ((august and December) and then divided this by the number of days since the previous harvest to give you the herbage accumulation rate (kg DM/ha per day). This needs to be clearly stated in this section. The LAR and SAR should then be clearly defined in a similar manner i.e. Proportion of leaf and stem were calculated by separating the 5 subsample tillers in to leaf, stem and dead and multiplied by fresh weight of herbage to give dry weight of leaf and stem.
L137-138: Do you need to define NDFap here? Were both NDF and ADF adjusted for ash and protein? I would suggest if both were, then you do not need to define NDFap, just use NDF and ADF. There is some inconsistency in the use of NDF and NDFap, for example in L161 and Equation 3, NDFap is used but in equation 4, NDF is used? Please be more consistent with the use of this terminology.
L162: Equation 4 – IN the paper of Sniffen et al. 1992 the equation for the C fraction (unavailable fiber) is CCj (%CH0) = 100 * NDFj (%DM) * 0.01 * LIGNINj (%NDF) *2.4) / CHOj (%DM). In your equation 4 you do not divide by the CHO %. Is there a reason for this? Also, recent research has demonstrated that the relationship between NDF and Lignin*2.4 is not reliable and that other approaches should be utilised to measure indigestible NDF (Reffrenato et al., 2018). Did you try to measure indigestible NDF in vitro or did you just use the Lignin * 2.4 calculation? I suggest that you clarify why you used the methodology you did. Also you need to mention the advances in the measurement of uNDF and that there may be some limitations to the methods that you have used here.
L177: Not sure about the journal specifications but generally you should not start a sentence with an abbreviation. Please spell out Herbage accumulation rate here. Also, please double check the number of decimal points required for P-values, personally I think 3 decimal points is enough.
L177-178: I find this sentence slightly confusing as irrigation is mentioned and then the genotype by season interaction is mentioned as being in Table 1. Irrigation is not included in Table 1 but the way the sentence is written it seems that it should be. If you want to talk about irrigation then I would suggest that you split this sentence into two. Talk about irrigation on its own and present the results for irrigation. Then you can go onto talk about the genotype by season interaction and mention Table 1. This way there is a clear differentiation between what results you are just presenting in the text and what results you are presenting in Table 1. This is a recurring issue for the way the results are presented in most of the tables and needs to be rectified in my opinion as it makes it difficult to read and interpret the results.
L185: Table 1 – The border lines of Table 1 are a bit inconsistent (i.e. different sizes and lengths), please amend accordingly. In Title “Herbage accumulation rate (HAR) and stem accumulation rate (SAR; kg dry matter/ha per day) as affected by the interaction between genotype and season….”. Link the footnote to the table using supersrcripts. Also, my preference for the footnote would be to explain the letter closest to the value (i.e. the capital letters first and then explain the letters further away from the value second (i.e. the lower case letters).
L192: “Mott had the lowest SAR”
L194-198 and L204-206 are similar and seem to say the same thing. Please combine or shorten to avoid repetition.
L200: Table 2 - Link the footnote to the table using supersrcripts. Also, my preference for the footnote would be to explain the letter closest to the value (i.e. the capital letters first and then explain the letters further away from the value second (i.e. the lower case letters).
L207: This is the frost mention of Leaf:stem ratio. Leaf:stem ratio should be defined in the materials and methods and the abbreviation L:S should also be shown in the materials and methods first.
L212: Table 3 - Link the footnote to the table using supersrcripts. Also, my preference for the footnote would be to explain the letter closest to the value (i.e. the capital letters first and then explain the letters further away from the value second (i.e. the lower case letters).
L220-221: From Table 4 it seems that Mott and Taiwan A-146 2.37 are not significantly different even though there is a numerical difference unless I am mistaken? If that is the case then, although true it is not technically correct to say that Mott had a 19% greater IVDDM than other genotypes. Please amend this sentence. Also Replace “Table 1” with “Table 4”.
L222-225: Greater ADF concentrations did not occur for irrigation in either the dry or the rainy season. Please amend this sentence and separate out the effects on ADF and NDF. “Greater NDF and concentrations occurred when elephantgrass genotypes were irrigated during the dry season and for irrigated and rainfed treatments during the rainy season compared to rainfed during the dry season (Table 4).”
L226: Table 4 - Link the footnote to the table using supersrcripts. Also, my preference for the footnote would be to explain the letter closest to the value (i.e. the capital letters first and then explain the letters further away from the value second (i.e. the lower case letters). Why does the order of the letters used change to denote significance change? For example in Table 4 for CP for Mott Aaα is used and for Taiwan A-146 2.37 Aabα is used. Whereas for NDF for Mott Abα is used and for Taiwan A-146 2.37 Aaα is used. You have changed the order of how you use the letters and I find it makes the tables very hard to follow. It is similar for ADF and IVDDM, in that both have a different lettering system for denoting significance. I know it is a small thing but it would make it easier for the reader if the same format was used for all tables.
L232: “Leaf lignin concentration (g/kg)…”
L236: Table 5 - Link the footnote to the table using supersrcripts. Also, my preference for the footnote would be to explain the letter closest to the value (i.e. the capital letters first and then explain the letters further away from the value second (i.e. the lower case letters).
L245: Replace “least” with “lowest”
L252: Table 6 – Same comment as for Table 4 regarding use of letters denoting significance.
L257-262: Again, this section is confusing to read, as it is not clear what you are referring to as being in the table and not in the table. I have rephrased this section below to show you how I think it should be written to make it clearer what is in the table or not. Also, what you are saying in the text does not match what is in the table unless I am not reading the Tables correctly? Irrigation increased NDF concentration only in the dry season not in the rainy season? Please amend the results section to reflect these comments. “Stem NDF concentration was affected by an irrigation × season interaction (P < 0.0001). Geater NDF concentration occurred for stems when elephantgrass was irrigated in the dry season compared to rainfed (Table 7), with averages of 706 and 649 g/kg DM, respectively. Genotype also had an affect on stem NDF concentration (P < 0.0001). Mott had the lowest NDF concentration (633 g/kg DM) and Taiwan A-146 2.37 showed intermediary values (693 g/kg DM). Elefante B and IRI 381, on the other hand, showed the greatest stem NDF concentration (720 and 709 g/kg DM), respectively.”
L264: Table 7 - Link the footnote to the table using supersrcripts. Also, my preference for the footnote would be to explain the letter closest to the value (i.e. the capital letters first and then explain the letters further away from the value second (i.e. the lower case letters).
L275: Table 8 - Link the footnote to the table using supersrcripts. Also, my preference for the footnote would be to explain the letter closest to the value (i.e. the capital letters first and then explain the letters further away from the value second (i.e. the lower case letters).
L281-282: Again, the text does not seem to match what is in the Table 9. The average Leaf of the N fraction A from the Table is 173 g/kg CP not 205 g/kg CP. Please amend this here and elsewhere in the text.
L282-284: Rephrase this sentence
L285: Table 9 - Link the footnote to the table using supersrcripts. Also, my preference for the footnote would be to explain the letter closest to the value (i.e. the capital letters first and then explain the letters further away from the value second (i.e. the lower case letters). Same comment as for Table 4 regarding use of letters denoting significance.
L304: Table 10 - Link the footnote to the table using supersrcripts. Also, my preference for the footnote would be to explain the letter closest to the value (i.e. the capital letters first and then explain the letters further away from the value second (i.e. the lower case letters). Same comment as for Table 4 regarding use of letters denoting significance.
L317: Table 11 - Link the footnote to the table using supersrcripts. Also, my preference for the footnote would be to explain the letter closest to the value (i.e. the capital letters first and then explain the letters further away from the value second (i.e. the lower case letters). Same comment as for Table 4 regarding use of letters denoting significance.
L325: Table 12 - Link the footnote to the table using supersrcripts. Also, my preference for the footnote would be to explain the letter closest to the value (i.e. the capital letters first and then explain the letters further away from the value second (i.e. the lower case letters). Same comment as for Table 4 regarding use of letters denoting significance.
L270-338: Please go through this section and amend how the results are written based on my previous comments. I am not going to go through this section in detail as it contains many of the same issues I have highlighted previously. Make sure that what you are writing matches what is in the Tables and avoid sentences whereby you refer to data that is in a Table and data that is just presented in the text.
L343-344: “This shows the influence of the ambient climate on the nutritive value and morphological aspects of forage.”
L352: “…greater potential for forage accumulation….”
L360: “contributes”
L362: “L:S ratio”
L364: I do not think that a sentence should start with a reference “[4]”. Please give the name of the first author e.g. “Carvalho et al. [4] reported…”. Amend elsewhere in the discussion also.
L370-373: I am not sure that this is a valid statement to include here. Although it true that a ruminal ammonia level of greater than 5 – 6 mg/dl is required in order to maintain adequate microbial growth, just because the diet is greater than 70 g/kg CP, this does not mean that there was adequate CP in the rumen or diet of the animal. Please rephrase this sentence or provide further evidence to support your point.
L374: Please spell out reference here to start the sentence.
Author Response

(The authors gave the same response as above.)
